# Aversive stimuli bias corticothalamic responses to motivationally significant cues

Federica Lucantonio[†], Eunyoung Kim, Zhixiao Su, Anna J Chang, Bilal A Bari, Jeremiah Y Cohen*

The Solomon H Snyder Department of Neuroscience, Brain Science Institute, Kavli Neuroscience Discovery Institute, The Johns Hopkins University School of Medicine, Baltimore, United States

**Abstract** Making predictions about future rewards or punishments is fundamental to adaptive behavior. These processes are influenced by prior experience. For example, prior exposure to aversive stimuli or stressors changes behavioral responses to negative- and positive-value predictive cues. Here, we demonstrate a role for medial prefrontal cortex (mPFC) neurons projecting to the paraventricular nucleus of the thalamus (PVT; mPFC→PVT) in this process. We found that a history of aversive stimuli negatively biased behavioral responses to motivationally relevant cues in mice and that this negative bias was associated with hyperactivity in mPFC→PVT neurons during exposure to those cues. Furthermore, artificially mimicking this hyperactive response with selective optogenetic excitation of the same pathway recapitulated the negative behavioral bias induced by aversive stimuli, whereas optogenetic inactivation of mPFC→PVT neurons prevented the development of the negative bias. Together, our results highlight how information flow within the mPFC→PVT circuit is critical for making predictions about motivationally-relevant outcomes as a function of prior experience.

*For correspondence:
jeremiah.cohen@jhmi.edu

Present address: [†]Department of Psychiatry and Taylor Family Institute for Innovative Psychiatric Research, Washington University School of Medicine, St Louis, United States

Competing interests: The authors declare that no competing interests exist.

## Introduction

Effective decision making requires anticipating the outcomes associated with environmental stimuli. It also requires balancing the goals of a decision—for instance, acquiring a reward or avoiding a punishment—with uncertainty about the outcome of the decision. For example, when the outcome of a decision to approach or avoid a stimulus is ambiguous, the nervous system must weigh the cost of receiving a punishment, or missing out on a reward, with the benefit of obtaining the reward, or avoiding the punishment.

Many decisions are influenced by background emotional state. For example, both positive and negative mood affect decision making (*Deldin and Levin, 1986*; *Wright and Bower, 1992*; *Bechara et al., 2000*; *Hockey et al., 2000*; *Dolan, 2002*; *Harding et al., 2004*). This background state can be driven by prior experience. For example, prior exposure to aversive stimuli or stressors changes behavioral responses to ambiguous stimuli (*Harding et al., 2004*; *Boleij et al., 2012*; *Rygula et al., 2014*). How the balance between competing behaviors is weighed in the brain or how prior experience with an environment shifts this balance is still poorly understood.

The medial prefrontal cortex (mPFC) is critical for regulating cue-mediated behaviors in both appetitive and aversive domains (*Ishikawa et al., 2008*; *Burgos-Robles et al., 2009*; *Sotres-Bayon and Quirk, 2010*; *Amemori and Graybiel, 2012*; *Burgos-Robles et al., 2013*; *Courtin et al., 2014*; *Sangha et al., 2014*; *Sparta et al., 2014*; *Burgos-Robles et al., 2017*; *Otis et al., 2017*). Behavioral manifestations of appetitive and aversive conditioning correlate with changes in neural activity within the mPFC (*Burgos-Robles et al., 2009*; *Peters et al., 2009*; *Amemori and Graybiel,*

*2012*; *Burgos-Robles et al., 2013*; *Moorman and Aston-Jones, 2015*) and pharmacological or optogenetic manipulations of mPFC alter both reward-seeking and fear-related behaviors (*Morgan and LeDoux, 1995*; *Blum et al., 2006*; *Corcoran and Quirk, 2007*; *Sierra-Mercado et al., 2011*; *Sangha et al., 2014*; *Sparta et al., 2014*; *Bari et al., 2019*).

The mPFC has dense projections to subcortical structures involved in motivated behavior, including the paraventricular thalamus (PVT) (*Vertes, 2004*; *Li and Kirouac, 2012*). Like the mPFC, PVT is recruited by cues or contexts previously associated with rewarding or aversive outcomes (*Beck and Fibiger, 1995*; *Schiltz et al., 2007*; *Yasoshima et al., 2007*; *Choi et al., 2010*; *Igelstrom et al., 2010*; *Haight and Flagel, 2014*; *Hsu et al., 2014*; *Do-Monte et al., 2015*; *Kirouac, 2015*; *Matzeu et al., 2015*; *Penzo et al., 2015*; *Li et al., 2016*; *Zhu et al., 2016*; *Do-Monte et al., 2017*; *McNally, 2021*). PVT neurons are activated by multiple forms of stressors (*Chastrette et al., 1991*; *Sharp et al., 1991*; *Cullinan et al., 1995*; *Bubser and Deutch, 1999*; *Spencer et al., 2004*) and coordinate behavioral responses to stress (*Hsu et al., 2014*; *Do-Monte et al., 2015*; *Penzo et al., 2015*; *Zhu et al., 2016*; *Do-Monte et al., 2017*; *Beas et al., 2018*). On the other hand, under conditions of opposing emotional valence, PVT plays a role in multiple forms of stimulus-reward learning and PVT neurons have been reported to show reward-modulated responses (*Schiltz et al., 2005*; *Igelstrom et al., 2010*; *Martin-Fardon and Boutrel, 2012*; *James and Dayas, 2013*; *Browning et al., 2014*; *Haight and Flagel, 2014*; *Li et al., 2016*; *Choi et al., 2019*). Recent work suggests that PVT is especially critical during motivational conflict (*Li et al., 2014*; *Choi and McNally, 2017*; *Zhu et al., 2018*; *Engelke et al., 2021*). Inhibition across the anterior–posterior axis of the PVT disrupts arbitration between appetitive and aversive behaviors when they are in conflict but has no effect when these behaviors are assessed in isolation (*Choi et al., 2019*). Moreover, activity in mPFC neurons projecting to the PVT also suppresses both the acquisition and expression of conditioned reward seeking (*Otis et al., 2017*).

Taken together, these studies place the mPFC-to-PVT projection in a position to integrate information about positive and negative motivationally relevant cues and translate it into adaptive behavioral responses (*McNally, 2021*). How these projection-specific mPFC neurons regulate behavioral responses in ambiguous settings and how their neural activity may be altered upon presentation of an aversive stimulus is unknown.

To address these questions, we trained mice on a go/no-go discrimination task with sweet- and bitter-predicting odor cues. Subsequently, mixtures of varying proportions of those cues were presented to probe behavioral responses to ambiguous stimuli. We recorded extracellularly from single neurons in mPFC and from identified mPFC→PVT neurons. We then tested whether stimulating or inhibiting mPFC→PVT projections modulated behavioral responses to the learned stimuli and their ambiguous mixtures.

## Results

### Aversive stimuli negatively bias behavioral responses to motivationally-significant cues

To assess the effect of aversive stimulus history on decisions about motivationally-significant outcomes, we developed a go/no-go discrimination task in head-fixed mice consisting of four phases: conditioning, probe test, reversal, and a second probe test (*Figure 1A*). In the conditioning phase, four odor cues (A, B, C, and D, counterbalanced) were presented. Odor A predicted an appetitive sweet solution (3 µl of 5% sucrose water). Odor B predicted an aversive bitter solution (3 µl of 10 mM denatonium water). Odor C was associated with no reinforcement. Odor D predicted a punishment (an unavoidable air puff delivered to the mouse's right eye) in mice assigned to the air-puff group and was associated with no reinforcement in mice assigned to the no-air-puff group. Each behavioral trial began with an odor (1 s; conditioned stimulus, CS), followed by a 1 s delay and an outcome (unconditioned stimulus, US). Mice showed essentially binary responses to cues that predicted sucrose or denatonium: they licked in anticipation of sucrose and did not in anticipation of denatonium (*Figure 1B,C*). We designed a probe test, in which, in addition to the four conditioning odors, we measured behavioral responses to ambiguous stimuli. We designed parametrically-varying mixtures of stimuli between appetitive and aversive solutions. We exposed mice to unreinforced mixtures of varying proportions of odors A and B: 85%A/15%B, 50%A/50%B, and 15%A/85%B. To

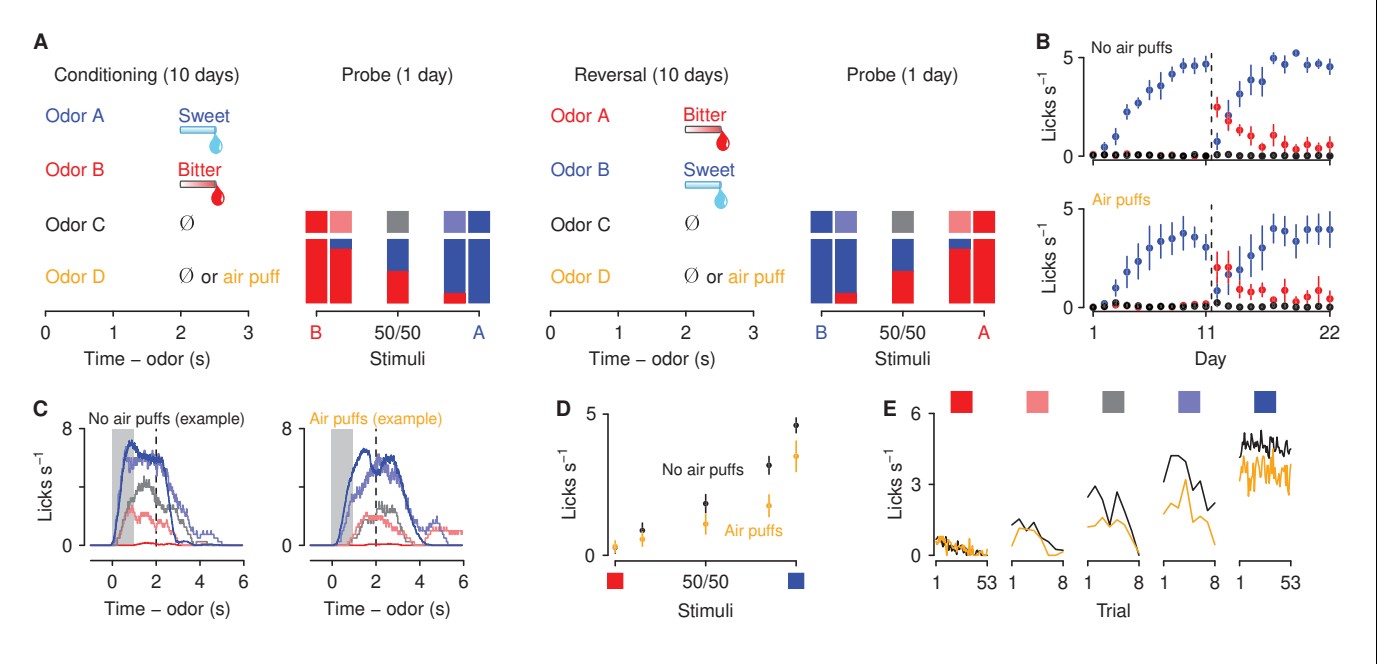

**Figure 1.** Behavioral responses to motivationally-significant predictive cues are modulated by history of aversive stimuli. (**A**) Task design and experimental timeline. During ten conditioning sessions, odors (A, B, C, and D) predicted an appetitive sucrose solution, an aversive denatonium solution, no outcome, and either no outcome or an unavoidable air puff, respectively. During the first probe test, A and B were mixed in three different ratios: 85%B/15%A (light red), 50%B/50%A (gray), 15%B/85%A (light blue). After completion of 10 reversal training sessions, in which A and B contingencies were reversed, mice were re-trained in a second probe test. (**B**) Licking rates in no air puff (top, 7 mice) or air puff (bottom, 5 mice) groups across days, during odor and delay period, for sucrose (blue), denatonium (red), and no-outcome (black) trials. Dashed lines indicate reversals on day 11. (**C**) Licking behavior from a representative test session from a mouse without (left) and one with (right) exposure to air puffs. Color gradations between blue and red indicate odor mixtures as in (**A**). Gray bars indicate a period of odor presentation. Dashed lines indicate outcome delivery. (**D**) Licking rates during sucrose (blue square) and denatonium (red square) trials and during the eight probe trials for no air puff (black) and air puff (orange) groups, during odor and delay period. (**E**) Mean trial-by-trial licking rates during sucrose (blue square) and denatonium (red square) trials and during the eight probe trials for each ambiguous cue (light red, gray, light blue squares) for no air puff (black) and air puff (orange) groups, during odor and delay period. Line and error bars represent mean ± SEM.

The online version of this article includes the following figure supplement(s) for figure 1:

**Figure supplement 1.** Mean ± SEM licking rates during anticipation periods (CS and delay), for mice exposed (orange) or unexposed (black) to air puffs.

ensure that behavioral responses to ambiguous stimuli were not driven by a mouse's preference for a particular odor, after completion of the probe test, cue-outcome associations for odors A and B were reversed, and each mouse was re-tested in a second set of probe stimuli.

As predicted, mice in both air-puff and no-air-puff groups quickly learned the CS-US associations: they showed increases in anticipatory licking responses to the positive, sucrose-predicting cue and in the delay before sucrose arrived across conditioning sessions, while withholding licking after sampling the negative, denatonium-predicting cue (*Figure 1B*). A three-factor ANOVA (session × cue × group) comparing licking during sucrose and denatonium cue presentation and delay period demonstrated a significant main effect of session ($F_{1,9} = 29.74$, $p<0.01$) and cue ($F_{1,1} = 64.77$, $p<0.01$). Moreover, mice exposed to air puffs responded to the sucrose-predicting odor with fewer licks (cue × group interaction, $F_{1,1} = 7.84$, $p<0.05$). During reversal learning, in which A and B contingencies were reversed, mice in both no-air-puff and air-puff groups quickly re-adapted to the new associations (*Figure 1B*). A three-factor ANOVA (session × cue × group) comparing licking rates during sucrose and denatonium cue presentation and delay period demonstrated a significant interaction between cue and session ($F_{1,9} = 8.4$, $p<0.01$).

At the end of conditioning and reversal training, mice received a single probe test session. Behavioral responses from the initial probe test were not statistically different from data gathered in the second test and thus these sessions were analyzed together in the text (*Figure 1—figure*

*supplement 1*). Licking rates for odor mixtures scaled with the proportion of the mixture that was the sucrose-predicting odor. This indicates that mice responded to parametrically varying ambiguous stimuli with smoothly varying behavioral responses (*Figure 1C*). Interestingly, mice exposed to air puffs responded to ambiguous odor mixtures with fewer licks during the anticipatory odor and delay period indicating that exposure to aversive stimuli biased decisions about ambiguous outcomes toward avoidance and away from approach (*Figure 1D*). A two-factor ANOVA (cue × group) comparing licking rates during cue presentation and delay period in test days demonstrated a significant interaction between cue and group ($F_{1,4} = 5.32$, $p<0.01$; Tukey HSD, $0.71 \pm 0.41$ licks s$^{-1}$, $p<0.001$). Lick rates were significantly larger for mixtures of odors in mice unexposed to air puffs ($t_{69} = 2.70$, $p<0.01$). Notably, the reduced licking to ambiguous cues in the air-puff group was evident on the first trial of the probe test and persisted throughout all probe test trials (*Figure 1E*). Thus, the decline in responding was not due to effects of extinction in the probe test. Indeed, both groups showed similar extinction of responding to ambiguous cues across trials resulting from outcome omission. A three-factor ANOVA (cue × trial × group) revealed a significant interaction between cue and trial ($F_{1,14} = 5.33$, $p<0.01$). Importantly, the interaction between cue, group and trial was not significant ($F_{1,14} = 1.21$, $p = 0.27$).

## Aversive stimuli modulate mPFC neuronal responses to motivationally relevant cues

Medial PFC (mPFC) is known to be involved in learning (*Holland and Gallagher, 2004*; *Luk and Wallis, 2009*; *Alexander and Brown, 2011*; *Del Arco et al., 2017*; *Otis et al., 2017*; *Orsini et al., 2018*), stress (*Wellman, 2001*; *Cook and Wellman, 2004*; *Radley et al., 2004*; *Radley et al., 2005*; *Liston et al., 2006*; *Radley et al., 2006*; *Cerqueira et al., 2007*; *Wei et al., 2007*; *Liu and Aghajanian, 2008*; *Radley et al., 2008*; *Goldwater et al., 2009*; *Yuen et al., 2012*; *Adhikari et al., 2015*), and uncertainty (*Ernst and Paulus, 2005*; *Opris and Bruce, 2005*; *Sugrue et al., 2005*; *Bach et al., 2009*; *Levy et al., 2010*; *Orsini et al., 2018*). These functions are critical for making predictions about previously unobserved stimuli. Such predictions derive from prior knowledge, as well as experience with the context of an environment. We thus asked whether mPFC neurons responded differently to motivationally relevant stimuli in the presence or absence of a history of aversive cues.

We recorded action potentials extracellularly from 2208 mPFC neurons in 12 mice, five exposed to air puffs (929 neurons), seven unexposed (1279 neurons), while mice performed the go/no-go discrimination task (*Figure 2—figure supplement 1*). Most neurons showed firing rate increases or decreases following odor cues, largely with persistent activity that lasted beyond cue offset (*Figure 2A*). There was a positive correlation between firing rates in response to sucrose-predicting and denatonium-predicting CS in mice exposed ($r = 0.59 \pm 0.04$, 95% CI, $p<0.0001$) and unexposed ($r = 0.58 \pm 0.04$, 95% CI, $p<0.0001$) to air puffs. To quantify the responses of the population, we measured the temporal response profile of each neuron during sucrose trials by quantifying firing rate changes relative to denatonium trials in 100 ms bins using a receiver operating characteristic (ROC) analysis (*Figure 2B*). We calculated the area under the ROC curve (auROC), comparing sucrose trials to denatonium trials. This analysis determines to what extent an ideal observer could discriminate between activities on the two trial types. Values of 0.5 indicate no discriminability, whereas values of 0 or 1 indicate perfect discriminability. A large fraction of neurons (270 of 929 in mice exposed to air puffs and 325 of 1279 in mice unexposed) showed auROC values greater than 0.7, indicating substantially greater activity for sucrose compared to denatonium trials, or values less than 0.3, indicating substantially greater activity for denatonium compared to sucrose trials (*Figure 2B*). These patterns were observed in mice exposed and unexposed to air puffs (*Figure 2C*). Neurons from mice exposed to air puffs responded robustly to both air puff-predicting cues and the air puffs themselves (*Figure 2D*; Wilcoxon rank sum test, $p<0.0001$).

We next asked whether a history of air puffs changed the responses of mPFC neurons to aversion- and reward-predictive cues together with ambiguous stimuli. We recorded from 136 neurons from mPFC in the no-air-puff group and 106 neurons from mPFC in air-puff exposed mice during the probe sessions (*Figure 3*). We calculated firing rates in response to odors predicting sucrose or denatonium, as well as the three proportions of mixtures of the two odors. These populations included 37 in the no-air-puff group and 25 in the air-puff group that exhibited auROC discrimination between sucrose and denatonium trials greater than 0.7 or less than 0.3 during the probe sessions (see *Figure 3A,B* for two example neurons in mice unexposed or exposed to air puffs). To

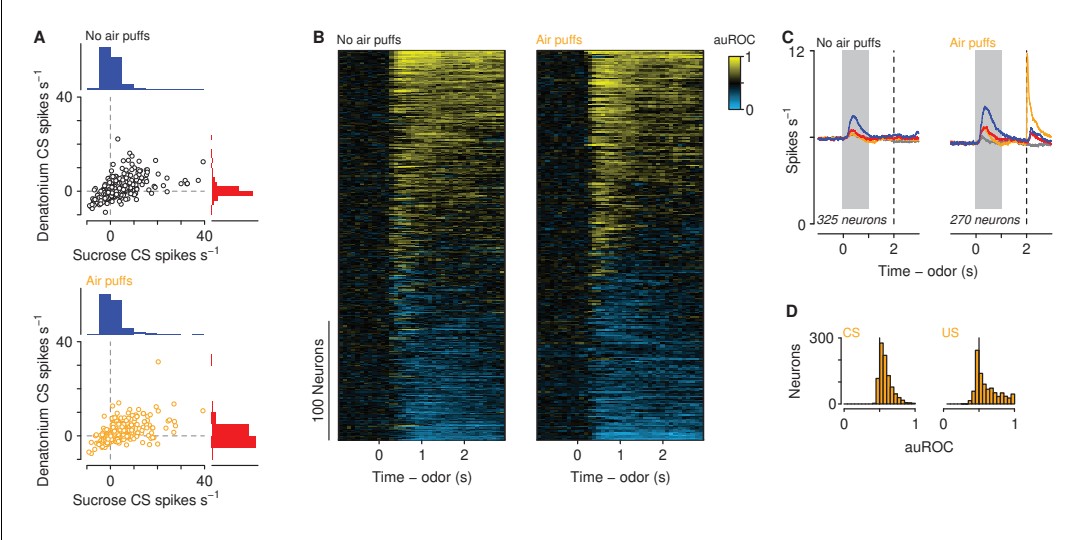

**Figure 2.** Neuronal responses across learning. (**A**) Firing rates of neurons in mice unexposed (top) and exposed (bottom) to air puffs relative to pre-CS firing rates. Histogram scale bars: 700 neurons (top), 500 neurons (bottom). (**B**) Discriminability (auROC) between sucrose and denatonium trials of neurons in mice unexposed (left) and exposed (right) to air puffs with significant firing rate changes during the cue. Increases (yellow) and decreases (cyan) in firing rate in sucrose trials relative to denatonium trials. Each row represents one neuron. (**C**) Average firing rates of all neurons with auROC values greater than 0.7 or less than 0.3 in at least one bin in no air puff (left) and air puff (right) mice during sucrose (blue), denatonium (red), no-outcome (gray) and air puff (orange) trials. Gray bars indicate a period of odor presentation. Dashed lines indicate outcome delivery. (**D**) auROC values for responses to air puff-predicting CSs (left) and air puff (right).

The online version of this article includes the following figure supplement(s) for figure 2:

**Figure supplement 1.** Drawings illustrate recording sites in mPFC in no air puff (left) and air puff exposed (right) mice.

compare firing rates of those neurons between mice exposed or unexposed to air puffs, we calculated a generalized linear model (Poisson regression) to predict spike counts during the cues as a function of air puff exposure and odor type. This type of regression tests for effects of categorical variables (i.e. air puff exposure) on variables generated from a discrete nonnegative probability distribution (i.e. spike counts). Cue-evoked firing rates were significantly higher in mice exposed to air puffs (*Figure 3C–E*; odor mixture $z = 0.18 \pm 0.015$, air-puff group $z = 0.043 \pm 0.014$, stimulus-group interaction $z = 0.033 \pm 0.019$, $p<0.001$). Firing rates were higher at each odor mixture in mice exposed to air puffs (Wilcoxon rank sum tests, all $p<0.05$). When we use the term odor mixture in the context, we refer to all mixtures of sucrose-predicting and denatonium-predicting odors: 100%/0%, 85%/15%, 50%/50%, 15%/85%, and 0%/100%.

## Aversive stimuli modulate corticothalamic neuronal responses to motivationally-significant cues

The neural data described above suggest that mPFC neurons that discriminate between sucrose and denatonium trials are modulated by a history of aversive stimuli. One of the major projection targets of the mPFC is the paraventricular nucleus of the thalamus (mPFC→PVT). Studies examining the role of PVT in regulating behavioral responses have found that PVT neurons are activated by cues or contexts previously associated with positive or negative emotional outcomes (*Beck and Fibiger, 1995*; *Schiltz et al., 2007*; *Yasoshima et al., 2007*; *Choi et al., 2010*; *Igelstrom et al., 2010*; *Haight and Flagel, 2014*; *Hsu et al., 2014*; *Do-Monte et al., 2015*; *Kirouac, 2015*; *Matzeu et al., 2015*; *Penzo et al., 2015*; *Li et al., 2016*; *Zhu et al., 2016*; *Do-Monte et al., 2017*). Additionally, activity in these corticothalamic neurons suppresses both the acquisition and expression of conditioned reward seeking (*Otis et al., 2017*). Thus, we hypothesized that mPFC→PVT neurons encode information about appetitive and aversive stimuli, and that this information is critical for weighing prior experience in those predictions.

To test this hypothesis, we performed projection-specific electrophysiological recordings from mPFC→PVT neurons while mice performed the go/no-go task. We expressed the light-gated ion

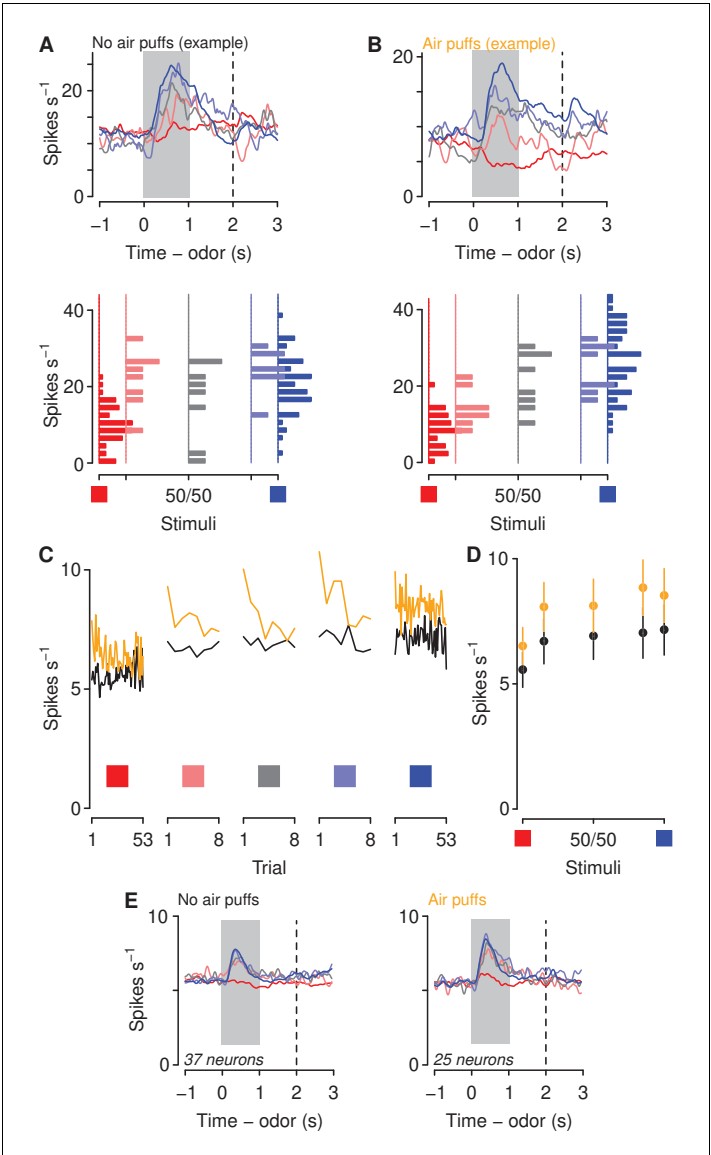

**Figure 3.** Aversive stimuli increase mPFC firing rates to motivationally relevant cues. (**A–B**) Average firing rates (top) and histograms of firing rates during odor and delay period (bottom) from example neurons in a mouse unexposed (**A**) and exposed (**B**) to air puffs. Red: denatonium trials. Blue: sucrose trials. Graded colors indicate mixtures as in *Figure 1A*. Gray bars indicate a period of odor presentation. Dashed lines indicate outcome delivery. (**C**) Mean trial-by-trial firing rates during sucrose (blue square) and denatonium (red square) trials and during the eight probe trials for each ambiguous cue (light red, gray, light blue squares) for no air puff (black) and air puff (orange) groups, during odor and delay period. Firing rates were calculated during the CS. (**D**) Mean ± SEM firing rates during sucrose (blue square) and denatonium (red square) trials and during the eight probe trials for no air puff (black) and air puff (orange) groups, during odor and delay period. (**E**) Mean firing rates of mPFC neurons in mice unexposed (left) or exposed to air puffs (right).

channel channelrhodopsin-2 (ChR2, using adeno-associated viruses, AAV1-CaMKIIa-ChR2-eYFP) in pyramidal neurons of the mPFC and we implanted an optic fiber above the posterior portion of the PVT, to activate mPFC→PVT cells antidromically (*Figure 4A,B*). Virus expression and optic fiber implantation were verified histologically (*Figure 4—figure supplement 1*). At the end of each recording session, we used ChR2 excitation to observe antidromically-evoked spikes. For each neuron, we measured the response to light stimulation and the shape of spontaneous spikes (*Figure 4C*). To unequivocally identify mPFC→PVT neurons, ChR2-expressing cells in the mPFC were

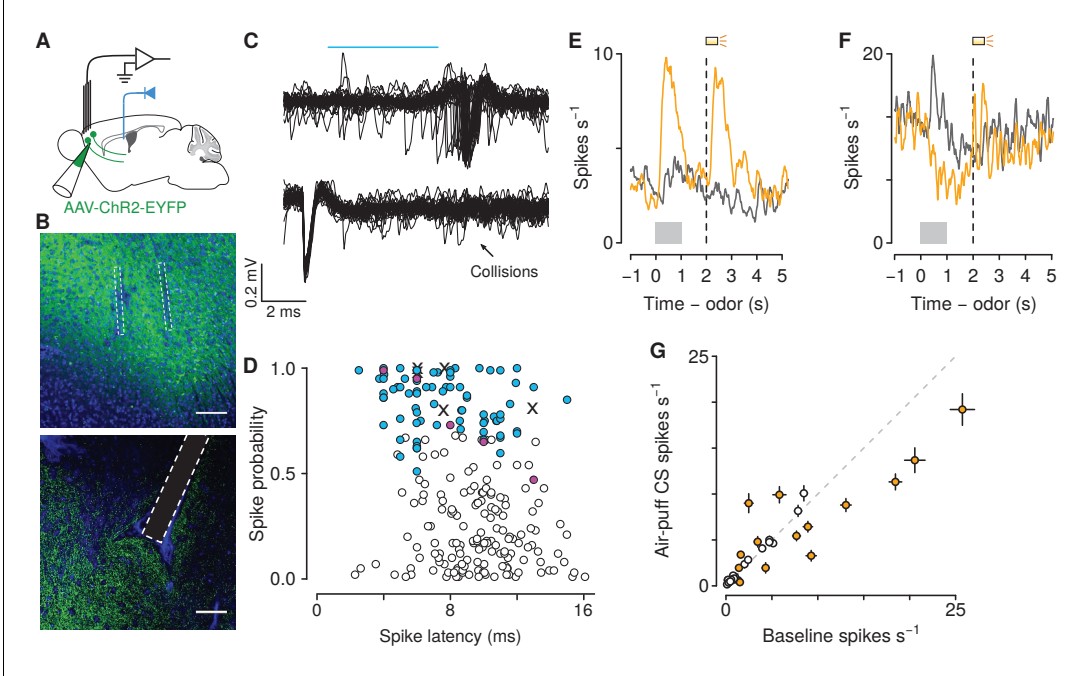

**Figure 4.** Air-puff-predicting stimuli modulate mPFC→PVT neuron firing rates. (**A**) Schematic drawings of viral stereotaxic injection of AAV1-CaMKII-ChR2-eYFP and tetrode bundle into mPFC and optic fiber over PVT. (**B**) eYFP (green), and DAPI (blue) in mPFC (top) and PVT (bottom) coronal sections from BL6 mice that received AAV1-CaMKII-ChR2-eYFP and tetrode bundle into mPFC and an optic fiber over PVT (scale bar, 100 μm). (**C**) Example of an identified corticothalamic neuron responding to a sequence of light stimuli (cyan) with action potentials (top) but not when the light stimuli followed spontaneous action potentials (bottom). (**D**) Antidromically-tagged corticothalamic neurons (blue) and antidromically-tagged corticothalamic neurons that passed collision tests (magenta). White points are neurons that were not identified. Crosses are neurons that passed collision tests, but were not recorded during behavior. (**E–F**) Average firing rates from example mPFC→PVT neurons showing firing rate increase (**E**) or decrease (**F**) to the air puff-predicting cue. Orange: air puff trials. Gray: CS - trials. Gray bars indicate odor presentation. Dashed lines indicate outcome delivery. (**G**) Scatter plot showing relationship between the change in firing rate to the air puff-predicting cue compared to baseline firing activity. Orange: neurons in which the firing rate during the air puff-predicting cue was significantly different from baseline firing activity (*t*-test, *p*<0.05).

The online version of this article includes the following figure supplement(s) for figure 4:

**Figure supplement 1.** Electrode and optic fiber locations.

identified with axonal photostimulation and extracellular recordings in mPFC using a collision test (*Figure 4D*; *Paintal, 1959*; *Bishop et al., 1962*; *Darian-Smith et al., 1963*; *Economo et al., 2018*; *Bari et al., 2019*). Based on the parameters of cells that passed the collision test, we only included units that responded to light with a latency less than 15 ms and spiked in response to at least 70% of all pulses (in response to 10 Hz pulses; *Figure 4D*). These criteria are comparable to the fastest responses seen using antidromic stimulation with collision tests in corticothalamic neurons in sensory regions of neocortex (*Swadlow and Weyand, 1981*; *Swadlow, 1998*; *Stoelzel et al., 2017*). Not every neuron was subjected to a collision test, therefore it is plausible that some of these neurons may have exhibited orthodromic activation via local synapses.

We identified 39 and 45 neurons as projecting to PVT in mice unexposed and exposed to air puffs, respectively (*Figure 4D*; no air puff group: 18 and 21 neurons during conditioning and probe sessions, respectively; air puff group: 35 and 10 neurons during conditioning and probe sessions, respectively). We first asked whether these neurons responded to aversive stimuli. Previous studies have emphasized the role of the PVT in shaping behavioral responses to stimuli that predict aversive outcomes (*Beck and Fibiger, 1995*; *Yasoshima et al., 2007*; *Hsu et al., 2014*; *Do-Monte et al., 2015*; *Penzo et al., 2015*), but it is unknown which of its inputs may drive those responses. We

found that 43% of mPFC→PVT neurons recorded from mice in the air puff group showed firing rate changes (5 of 35 neurons with increases and 10 of 35 neurons with decreases) in response to air-puff-predicting stimuli during conditioning (*Figure 4E–G*). This demonstrates that mPFC→PVT neurons, thought to be involved in behavioral responses to aversive-predicting stimuli, are indeed modulated by those stimuli.

We next compared the responses of mPFC→PVT neurons to sucrose- and denatonium-predictive cues and ambiguous stimuli (see *Figure 5A,B* for two example neurons in mice unexposed or exposed to air puffs, respectively). Interestingly, corticothalamic neurons from mice exposed to air puffs responded to the sucrose-predictive cue and ambiguous odor mixtures with higher phasic activity during the anticipatory odor and delay period, indicating that exposure to aversive stimuli biases neural responses of corticothalamic neurons to reward-predictive and ambiguous stimuli (*Figure 5C–F*; Poisson regression, odor mixture $z = 0.26 \pm 0.10$, $p<0.01$, air-puff group $z = 0.73 \pm 0.15$, $p<0.01$, stimulus-group interaction $z = 0.27 \pm 0.22$, $p>0.2$). Firing rates were higher at each odor mixture in mice exposed to air puffs (Wilcoxon rank sum tests, all $p<0.05$).

## Excitation of corticothalamic neurons modulates behavioral responses to motivationally-relevant cues

The neural data described above suggest that elevated activity in corticothalamic neurons to the aversion- and reward-predictive cues together with the ambiguous cues is critical for modulating behavioral response to those stimuli. To test this hypothesis, we next used optogenetic methods to activate corticothalamic neurons selectively at the time of presentation of those cues in mice trained in the same go/no-go task described above but with no exposure to air puffs. Mice received bilateral infusions of either AAV1/5-CaMKIIa-ChR2-EYFP or AAV1/5-CaMKIIa-EYFP (control) into mPFC;

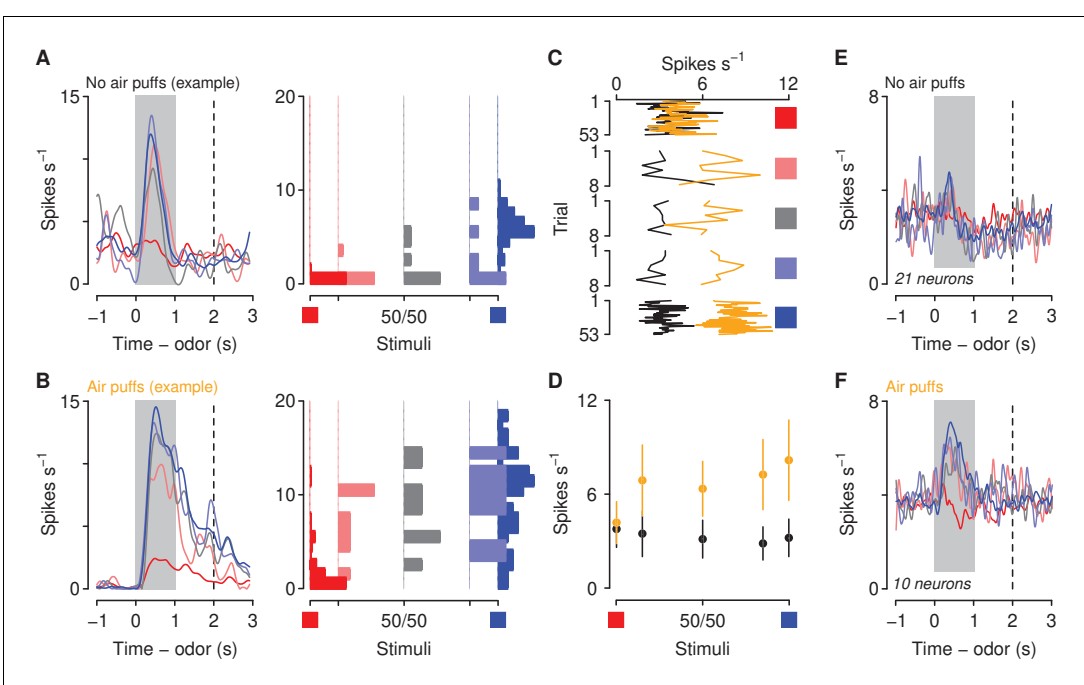

**Figure 5.** Aversive stimuli increase corticothalamic firing rates to motivationally-relevant cues. (**A–B**) Average firing rates (left) and histograms of firing rates during odor and delay period (right) from example neurons in a mouse unexposed (**A**) and exposed (**B**) to air puffs. Red: denatonium trials. Blue: sucrose trials. Graded colors indicate mixtures as in *Figure 1A*. Gray bars indicate a period of odor presentation. Dashed lines indicate outcome delivery. (**C**) Mean trial-by-trial firing rates during sucrose (blue square) and denatonium (red square) trials and during the eight probe trials for each ambiguous cue (light red, gray, light blue squares) for no air puff (black) and air puff (orange) groups, during odor and delay period. (**D**) Mean ± SEM firing rates during sucrose (blue square) and denatonium (red square) trials and during the eight probe trials for no air puff (black) and air puff (orange) groups, during odor and delay period. (**E**) Mean firing rates of corticothalamic neurons in mice unexposed to air puffs. (**F**) Mean firing rates of corticothalamic neurons in mice exposed to air puffs.

expression was verified histologically (*Figure 6—figure supplement 1*). Mice also received fiber optic implants over posterior PVT (*Figure 6A*). Three weeks after surgery, these mice began training in the go/no-go discrimination task and, during the probe session, light was delivered into the PVT in five random trials of all sucrose and denatonium trials and in half of all the ambiguous trials, in the cue and delay period (*Figure 6B*). We used a 10 Hz frequency stimulation because it approximated the firing rate of corticothalamic neurons in response to ambiguous stimuli observed in neural recordings; we also used a 20 Hz frequency in order to obtain a frequency response curve. All mice

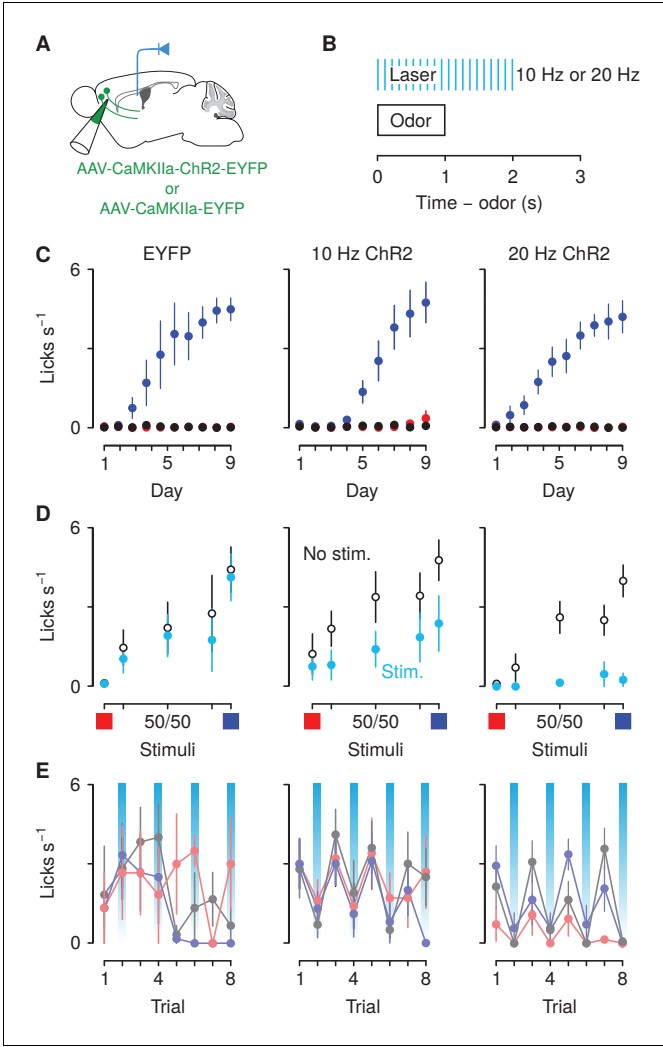

**Figure 6.** Optogenetic excitation of corticothalamic neurons negatively biases responses to motivationally-relevant stimuli. (**A**) Schematic of viral stereotaxic injection of AAV1/5-CaMKIIa-ChR2-eYFP or AAV1/5-CaMKIIa-eYFP into mPFC and optic fiber over PVT. (**B**) Optical stimulation was delivered during presentation of the cue and during the 1 s delay before outcome delivery. (**C**) Licking rates in eYFP (left, $n = 3$), 10 Hz ChR2-eYFP (center, $n = 5$), and 20 Hz ChR2-eYFP (right, $n = 7$) groups across conditioning, during odor and delay period, for sucrose (blue), denatonium (red), and no-outcome (black) trials. (**D**) Licking rates during sucrose (blue square) and denatonium (red square) trials and during the eight probe trials of each mixed stimulus for eYFP (left), 10 Hz ChR2-eYFP (center) and 20 Hz ChR2-eYFP (right) groups, during odor and delay period, with (light blue) or without (white) laser stimulation. (**E**) Trial-by-trial licking rates during 85%A/15%B (light red), 50%A/50%B (gray), 15%A/85%B (light blue) trials for eYFP (left), 10 Hz ChR2-eYFP (center) and 20 Hz ChR2-eYFP (right) groups, during odor and delay period, with (light blue shadows) or without laser stimulation. Line and error bars represent the mean ± SEM. The online version of this article includes the following figure supplement(s) for figure 6:

**Figure supplement 1.** Virus expression and optic fiber locations.

**Figure supplement 2.** mPFC→PVT stimulation did not suppress licking for unexpected rewards.

underwent conditioning sessions and there were neither main effects nor any interactions of group on conditioned responding across conditioning ($F<1.14$; $p>0.33$; **Figure 6C**). During the subsequent probe test, ChR2 mice showed a reduction in response to denatonium- and sucrose-predictive cues and ambiguous cues in the trials in which light was delivered in a frequency-dependent manner, whereas eYFP mice that received the same treatment responded equally in trials with or without light delivery (**Figure 6D,E**). A three-factor ANOVA (cue × stimulation × group) comparing licking during cue presentation and delay period in stimulated versus unstimulated trials in EYFP, 10 Hz and 20 Hz groups revealed a significant main effect of cue ($F_{1,4} = 13.8$, $p<0.01$) and stimulation ($F_{1,1} = 30.11$, $p<0.01$). Moreover, there was a significant interaction between cue and group ($F_{1,8} = 4.8$, $p<0.01$) and stimulation and group ($F_{1,2} = 4.68$, $p<0.05$).

Stimulation of the same neurons did not, however, disrupt licking for an unpredictable reward. Outside of the task, we delivered randomly-timed sucrose rewards (3 µl). Mice licked at high rates to consume unexpected rewards (**Figure 6—figure supplement 2**). Importantly, stimulation of mPFC→PVT axons during reward delivery at the same frequencies used in the behavioral task in half of the total number of trials did not change the number of licks in response to these rewards. A two-factor ANOVA (frequency of stimulation × group) comparing licking during stimulated and unstimulated trials revealed no significant main effect or interaction with group (all $F<0.23$, $p>0.65$). Thus, uncued licking is not altered by optogenetic excitation of mPFC→PVT cells and the optogenetic effects are not due to a light-induced impairment in licking in general.

## Inhibition of corticothalamic neurons prevents negative biases to motivationally-relevant cues

The optogenetic data described above suggest that artificially increasing activity in corticothalamic neurons to the aversion- and reward-predictive cues together with the ambiguous cues is sufficient for modulating behavioral response to those stimuli. To test the necessity of this pathway in generating negative biases to motivationally relevant stimuli, we next used optogenetic inhibition to inhibit corticothalamic neurons selectively during air-puff trials during conditioning in mice trained in the same go/no-go task described above. Mice received bilateral infusions of either AAV1-CaMKIIa-eNpHR3.0-EYFP or AAV1-CaMKIIa-EYFP (control) into mPFC; expression was verified histologically (**Figure 6—figure supplement 1**). Mice also received fiber optic implants over the posterior portion of PVT (**Figure 7A**). Three weeks after surgery, these mice began training in the go/no-go discrimination task and, during the air-puff-predictive cue, light was delivered into the PVT (**Figure 7B**). All mice underwent conditioning sessions and there were main effects of odor ($F_{2,228} = 124.7$, $p<0.0001$) and day ($F_{1,228} = 205.5$, $p<0.0001$). During the late phase of training, licking rates were not significantly different between the two groups of mice (Wilcoxon rank sum test, $p>0.05$; **Figure 7C**). During the probe test, mice expressing EYFP showed a reduction in response to denatonium- and sucrose-predictive cues and ambiguous cues in the trials compared to mice expressing eNpHR3.0-EYFP (**Figure 7D**). A two-factor ANOVA (cue × group) comparing licking during cue presentation and delay period in eYFP versus virus groups revealed a significant main effect of cue ($F_{1,4} = 39.8$, $p<0.05$) and group ($F_{1,1} = 16.6$, $p<0.05$).

## Discussion

In this study, we examined the projection from mPFC to PVT in mice approaching or avoiding negative and positive valence-predictive stimuli. We found that a history of aversive stimuli negatively biased behavioral responses to those motivationally relevant cues while increasing excitatory responses of PVT-projecting mPFC neurons. Indeed, mice exposed to aversive stimuli showed reduced approach behavior to reward-predictive and ambiguous stimuli compared to mice unexposed to aversive stimuli. Moreover, artificially increasing activity from mPFC to PVT quantitatively mimicked the negative behavioral bias induced by presentation of aversive stimuli while selectively inhibiting this pathway during learning prevented the formation of negative biases.

Cognitive processes—appraisals of stimuli, events and situations—play an important role in generating affective states, and these affective states influence cognitive functioning by inducing attentional, memory, and judgment biases (**Lerner and Keltner, 2000**; **Haselton et al., 2009**; **Harding et al., 2004**; **Enkel et al., 2010**; **Rygula et al., 2012**; **Papciak et al., 2013**; **Rygula et al., 2013**; **Parker et al., 2014**; **Rygula et al., 2014**). Among brain regions that are engaged in affective

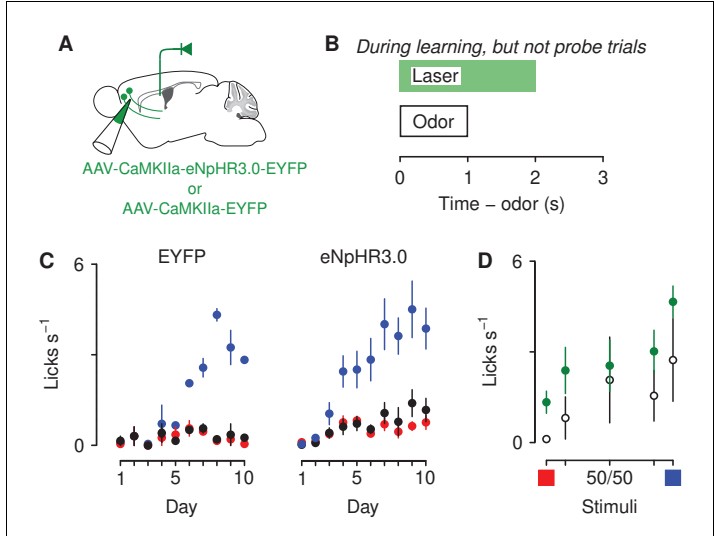

**Figure 7.** Optogenetic inhibition of corticothalamic neurons prevents negatively bias responses to motivationally-relevant stimuli. (A) Schematic of viral stereotaxic injection of AAV1-CaMKIIa-eNpHR3.0-EYFP or AAV1-CaMKIIa-EYFP into mPFC and optic fiber over PVT. (B) Optical stimulation was delivered during presentation of the air-puff-predicting cue, during the 1 s delay before air-puff delivery and during air-puff delivery, only during learning. (C) Licking rates in EYFP (left, $n = 3$), eNpHR3.0-eYFP (right, $n = 5$) groups across conditioning, during odor and delay period (sucrose: blue, denatonium: red, no-outcome: black). (D) Licking rates during sucrose (blue square) and denatonium (red square) trials and during the eight probe trials of each mixed stimulus for EYFP (white circles), eNpHR3.0-EYFP (green circles) groups, during odor and delay period. Line and error bars represent the mean ± SEM.

processing, the mPFC has long been implicated in adaptive responding by signaling information about expected outcome and by regulating sensitivity to reward and punishment (*Holland and Gallagher, 2004*; *Luk and Wallis, 2009*; *Alexander and Brown, 2011*; *Del Arco et al., 2017*; *Orsini et al., 2018*). Here, we found that firing rates in mPFC neurons reflect cue-evoked expectations for aversion- and reward-predictive cues and were enhanced in mice exposed to aversive stimuli, which correlates with a reduction in anticipatory responses to the same stimuli. These observations are consistent with a negative bias and suggest that negative events can bias decisions by altering the activity of mPFC neurons. Several studies have implicated the prefrontal network in the pathophysiology of affective disorders (*Phillips et al., 2003*; *Drevets et al., 2008*) and chronic stress—a crucial factor in increasing the risk of developing affective disorders—has profound detrimental effects on the anatomy and physiology of mPFC neurons (*Wellman, 2001*; *Cook and Wellman, 2004*; *Radley et al., 2004*; *Liston et al., 2006*; *Radley et al., 2006*; *Cerqueira et al., 2007*; *Wei et al., 2007*; *Liu and Aghajanian, 2008*; *Radley et al., 2008*; *Goldwater et al., 2009*; *Yuen et al., 2012*; *Adhikari et al., 2015*). For example, chronic stress induces significant regression of the apical dendrites of pyramidal neurons in mPFC (*Cook and Wellman, 2004*; *Radley et al., 2004*; *Liston et al., 2006*; *Goldwater et al., 2009*), which may in turn impact mPFC function.

The mPFC densely projects to subcortical structures relevant for our behavioral task, including PVT, amygdala, hippocampus, and nucleus accumbens (*Vertes, 2004*; *Li and Kirouac, 2012*). Among all brain regions that receive strong projections from the mPFC, PVT has long been considered a stress detector and implicated in the emergence of adaptive responding to stress (*Chastrette et al., 1991*; *Sharp et al., 1991*; *Cullinan et al., 1995*; *Bubser and Deutch, 1999*; *Spencer et al., 2004*; *Hsu et al., 2014*; *Do-Monte et al., 2015*; *Penzo et al., 2015*; *Zhu et al., 2016*; *Do-Monte et al., 2017*; *Beas et al., 2018*; *Choi et al., 2019*). At the same time, PVT has also been considered a potential mediator of motivated behavior responding to both food- and drug-associated cues (*Schiltz et al., 2005*; *Igelstrom et al., 2010*; *Martin-Fardon and Boutrel, 2012*; *James and Dayas, 2013*; *Browning et al., 2014*; *Haight and Flagel, 2014*; *Li et al., 2016*). Consistent with these findings that place PVT in a position to integrate information about positive and

negative motivationally relevant cues and translate it into adaptive behavioral responses (*McNally, 2021*), we found that mPFC neurons projecting to the PVT maintain cue-evoked expectations for motivationally-relevant outcomes and their neural activity was enhanced in mice exposed to aversive stimuli. A feature of our experimental design is worth comment. In our experiments, mice were exposed to aversive stimuli in the same context in which they received motivationally-relevant cues. This design cannot rule out the influence of context-derived motivational conflict on the observed behavioral outcome. Indeed, recent work suggests that PVT is important during motivational conflict (*Li et al., 2014*; *Choi and McNally, 2017*; *Zhu et al., 2018*; *Choi et al., 2019*; *Engelke et al., 2021*). PVT seems important for behavioral control when appetitive and aversive behaviors are elicited at the same time and the animal must select between them (*Li et al., 2014*; *Choi and McNally, 2017*; *Zhu et al., 2018*; *Choi et al., 2019*). Future studies in which aversive stimuli and rewarding stimuli are presented in two different contexts will help clarify the behavioral relevance of such context-derived variables.

By mimicking the increased response of mPFC→PVT neurons induced by exposure to aversive stimuli, we observed a reduction in anticipatory responses to the predictive cues. By contrast, selective inhibition of this pathway prevented the formation of negative biases. These findings suggest that information about cue interpretation is transferred from mPFC to PVT and this pathway is crucial for adaptive responding toward those stimuli as a function of previous experience. These results are also consistent with a recent study showing that activity in mPFC neurons projecting to the PVT suppresses both the acquisition and expression of conditioned reward seeking (*Otis et al., 2017*). Based on recent evidence showing two genetically, anatomically, and functionally distinct cell types across the anteroposterior axis of the PVT (*Gao et al., 2020*), it will be interesting to investigate in future studies whether the information from mPFC is transferred to anatomically or molecularly segregated cell types or projection-specific neurons within the PVT. Indeed, signals from mPFC to PVT inputs have been recently shown to propagate onto PVT projections to nucleus accumbens neurons (*Otis et al., 2019*) and are responsible for influencing reward seeking by adjusting a multiplexed reward signal in downstream PVT→nucleus accumbens neurons. Future studies may help clarify whether this pathway is also involved in negative bias or dynamic salience processing (*Zhu et al., 2018*), which is a behavioral domain that is well positioned to shape the negative bias itself.

Studying the mechanisms by which neuronal responses to cues are updated as a function of prior experience in the mPFC→PVT circuit is crucial for understanding learning and decision making more generally. Adaptive tuning of this network is modulated by ascending monoaminergic systems (*Arnsten et al., 2012*) and maladaptive changes of these systems has been implicated in cognitive deficits associated with several affective disorders (*Enkel et al., 2010*; *Kukolja et al., 2008*). For example, acute pharmacological stimulation of the serotonergic and dopaminergic systems has been shown to influence cognitive bias in rodents (*Rygula et al., 2014*). In particular, citalopram, a selective serotonin reuptake inhibitor, and amphetamine, a powerful psychostimulant, both induced a positive cognitive processing bias (*Rygula et al., 2014*). Those results are important from a clinical point of view, knowing that negative cognitive bias lies at the core of the pathophysiology of several affective disorders and it has been extensively studied in humans (*Wright and Bower, 1992*; *MacLeod and Byrne, 1996*; *Beck, 2008*). For example, it has been shown that patients with anxiety and depression interpret ambiguous information with a negative bias (*Schwarz and Clore, 1983*; *Eysenck et al., 1987*; *Wright and Bower, 1992*; *MacLeod and Byrne, 1996*; *Lawson et al., 2002*; *Beck, 2008*; *Chan et al., 2008*; *Pizzagalli et al., 2008*; *Dearing and Gotlib, 2009*). Thus, understanding how cognitive biases develop and act in several chronic and debilitating neuropsychiatric disorders may offer an opportunity for designing novel treatments, aimed at ameliorating the proper functional tuning and connectivity of prefrontal-orchestrated neuronal circuits.

In summary, by using exposure to aversive stimuli, we induced a negative bias in mice in response to ambiguous stimuli, which was linked to an increase in firing rates of PVT-projecting mPFC neurons. Artificial activation of the same pathway recapitulated the behavioral outcome while inhibition prevented the formation of negative biases. Thus, our results highlight a fundamental role for the mPFC→PVT circuit in shaping adaptive responses by modulating predictions about imminent motivationally-relevant outcomes as a function of prior experience.

## Materials and methods

### Subjects

Wild-type C57BL/6J male mice (The Jackson Laboratory, 000664), 8–10 weeks old at the time of surgery, were housed in a reverse 12 hr light-dark cycle room (lights on at 20:00). All mice were given ad libitum water except during testing periods. During behavioral testing, mice were water deprived by giving 1 ml of water per day. Food was freely available throughout the experiments. All testing was conducted in accordance with the National Institutes of Health Guide for the Care and Use of Laboratory Animals and approved by the Johns Hopkins University Animal Care and Use Committee (protocol MO19M424).

### Stereotaxic surgeries

All mice were surgically implanted with custom-made titanium head plates using dental adhesive (C and B-Metabond, Parkell) under isoflurane anesthesia (1.0–1.5% in $O_2$). Surgeries were conducted under aseptic conditions and analgesia (ketoprofen, 5 mg/kg and buprenorphine, 0.05–0.1 mg/kg) was administered postoperatively. Mice recovered for 7–10 days before starting behavioral testing.

For electrophysiological experiments, we implanted unilaterally a custom microdrive containing eight drivable tetrodes made from nichrome wire (PX000004, Sandvik) and positioned inside 39 ga polyimide guide tubes. We targeted mPFC under stereotaxic guidance at 2.2 mm anterior and 0.4 mm lateral to bregma and 1.6 mm ventral to the skull. Tetrodes were advanced subsequently into final positions in mPFC during recording. For identifying corticothalamic neurons, an optic fiber was implanted over PVT under stereotaxic guidance at 1.4 mm anterior and 1.3 mm lateral to bregma and 3.6 mm ventral to the skull with a 225° angle.

For optogenetic excitation experiments, AAV1/5-CamKIIa-hChR2(H134R)-eYFP or AAV1/5-CamKIIa-eYFP (AAV5: from UNC GTC Vectore Core; AAV1: from Addgene) was injected bilaterally in mPFC under stereotaxic guidance at 2.2 mm anterior and 0.3 mm lateral to bregma and 1.6 mm ventral to the skull. pAAV-CaMKIIa-hChR2(H134R)-EYFP was a gift from Karl Deisseroth (Addgene viral prep 26969-AAV1; http://n2t.net/addgene:26969; RRID:Addgene_26969) (*Lee et al., 2010*). A total of 300 nl of virus (titer ~ $10^{13}$ GC/mL) per hemisphere was delivered at the rate of 1 nl/s (MMO-220A, Narishige). The injection pipette was left in place for 5 min after each injection. Optic fibers (200 μm diameter, 0.39 NA, Thorlabs) were implanted bilaterally over mPFC (at 2.2 mm anterior and 0.6 mm lateral to bregma and 1.3 mm ventral to the skull with a 10° angle) or unilaterally over PVT (at 1.4 mm anterior and 1.3 mm lateral to bregma and 3.6 mm ventral to the skull with a 22.5° angle). For inhibition experiments, AAV1-CaMKIIa-eNpHR 3.0-EYFP or AAV1-CaMKIIa-EYFP was injected as described above. pAAV-CaMKIIa-eNpHR 3.0-EYFP was a gift from Karl Deisseroth (Addgene viral prep 26971-AAV1; http://n2t.net/addgene:26971; RRID:Addgene_26971).

### Behavioral task

Following recovery from surgery, water-restricted mice were habituated for 3 days while head-fixed before training on the go/no-go task. Each mouse performed behavioral tasks at the same time of day (between 08:00 a.m. and 2:00 p.m). All behavioral tasks were performed in dark, sound-attenuated chamber, with white noise delivered between 2 and 60 kHz (L60 Ultrasound Speaker, Pettersson). Odors were delivered with a custom-made olfactometer (*Cohen et al., 2012*). Each odor was dissolved in mineral oil at 1:10 dilution. Diluted odors (30 μl) were placed on filter-paper housing (Whatman, 2.7 μm pore size). Odorized air was further diluted with filtered air by 1:10 to produce a 1.0 l/min flow rate. Licks were detected by charging a capacitor (MPR121QR2, Freescale). Task events were controlled with a microcontroller (ATmega16U2 or ATmega328). Reinforcements were 3 μl of sucrose (an appetitive sweet solution), denatonium (an aversive bitter solution) or air puff (40 psi), delivered using solenoids (LHDA1233115H, The Lee Co). Intertrial intervals (ITIs) were drawn from an exponential distribution with a rate parameter of 0.3, with a maximum of 30 s. This resulted in a flat ITI hazard function, ensuring that expectation about the start of the next trial did not increase over time (*Luce, 1986*). The mean ITI was 7.2 s (range 2.4–30.0 s).

## Mice underwent 10 conditioning sessions

In each session, mice received 50 1 s presentations of four different olfactory stimuli (A, B, C, and D). The order of odor presentations was randomized among mice and among sessions. For all conditioning, A, B, C, and D consisted of (+)-limonene, p-cymene, penthylacetate, and acetophenone, respectively (counterbalanced). 1 s after termination of A, sucrose was delivered and 1 s after termination of B, denatonium was delivered. C was paired with no reinforcement. In the air puff group, 1 s after termination of D, an unavoidable air puff was delivered to their right eye, while in the control group, D was paired with no reinforcement. Four s after the presentation of each odor, a vacuum was activated to remove any residual of sucrose or denatonium. After the activation of the vacuum, there was a fixed 3 s delay and then the variable ITI will follow. After completion of conditioning training, mice received a single extinction probe session. During the probe session, the four conditioning odors were continued to be presented (53 trials for each odor), but mice also received eight non-reinforced presentations of three mixtures of varying proportions of A and B odors: 85%A/15%B, 50%A/50%B, 15%A/85%B. These odor mixture trials were interleaved with the four conditioning odor trials in a randomized order.

In mice designated for the mPFC electrophysiological experiments, following the probe test, mice underwent reversal learning, in which A and B were reversed. 1 s after termination of A, denatonium was delivered and 1 s after termination of B, sucrose was delivered. C and D were continued to be presented as in conditioning. Then, mice received another single extinction probe session, identical to the one received after conditioning. Neural data from the initial extinction days were not statistically different from data gathered in later rounds of training and thus these neurons were analyzed together in the text.

In mice designated for the PVT-projecting mPFC electrophysiological experiments, following the probe test, mice repeated 3 days of conditioning and then underwent additional rounds of probe test days in order to acquire additional data. This was done up to three times for a given mouse. Neural data from the initial extinction days were not statistically different from data gathered in later rounds of training and thus these neurons were analyzed together in the text.

In mice designated for the optogenetic experiments of stimulation, training began approximately 3 weeks after viral injection and fiber implantation, and light (473 nm, 10–12 mW) was delivered into the PVT during the probe session. During the behavioral task, light was delivered in half of all the ambiguous trials, during the cue and delay epoch. Moreover, light was also delivered in five random trials of all sucrose and denatonium trials, during the cue and delay epoch. The primary measure of conditioning to cues was the number of licks during odor presentation and the second preceding reinforcement delivery. During the uncued stimulation trials, light was delivered in half of all trials, during the presentation of reward delivery and lasted for 1500 ms.

In mice designated for the optogenetic experiments of inhibition, training began approximately 3 weeks after viral injection and fiber implantation, and light (532 nm, 10–12 mW) was delivered into the PVT during air puff trials during conditioning days. During the behavioral task, light was delivered in all air puff trials, during the cue and delay epoch. The primary measure of conditioning to cues was the number of licks during odor presentation and the second preceding reinforcement delivery.

## Electrophysiology

Throughout the discrimination task, mice were attached to the recording cable and before each session, tetrodes were screened for activity. Active tetrodes were selected for recording, and the session was begun. On the rare occasion that fewer than four of eight tetrodes had single units, the tetrode assembly was advanced 40 or 80 μm at the end of the session. Otherwise, the tetrode assembly was kept in the same position between sessions until the probe test day. A neuron may be represented more than one time in the dataset. After the extinction probe test, the tetrode assembly was advanced 80 μm regardless of the number of active tetrodes in order to acquire activity from a new group of neurons in any subsequent training.

We recorded extracellularly (Digital Lynx 4SX, Neuralynx Inc) from multiple neurons simultaneously at 32 kHz using custom-built screw-driven microdrives with eight tetrodes coupled to a 200 μm fiber optic (32 channels total). All tetrodes were gold-plated to an impedance of 200–300 kΩ prior to implantation. Spikes were bandpass filtered between 0.3–6 kHz and sorted online and

offline using Spikesort 3D (Neuralynx Inc) and custom software written in MATLAB. To measure isolation quality of individual units, we calculated the L-ratio (*Schmitzer-Torbert et al., 2005*) and fraction of interspike interval (ISI) violations within a 2 ms refractory period. All single units included in the dataset had an L-ratio less than 0.05 and less than 0.1% ISI violations. We only included units that had a firing rate of greater than 0.5 spikes s$^{-1}$ over the course of the recording session.

## Optogenetic identification

To verify that our recordings targeted corticothalamic neurons, at the end of daily recording sessions, we used channelrhodpsin excitation to observe stimulation-locked spikes, by delivering 3–5 ms pulses of 473 nm light at 15 mW using a diode-pumped solid-state laser (Laserglow), together with a shutter (Uniblitz). Spike shape was measured using a broadband signal (0.1 Hz-9 kHz) sampled at 32 kHz. This ensured that particular features of the spike waveform were not missed. We delivered 10 trains of light (10 pulses per train, 10 s between trains) at 10 Hz, resulting in 100 total pulses. For collision tests, we delivered light over PVT (2 ms pulse, 473 nm, 15 mW) triggered by a spontaneous action potential. Identified neurons did not show stimulus-locked spikes following spontaneous spikes ('collisions').

## Histology

At the end of behavioral testing, all mice were deeply anesthetized and then transcardially perfused with 4% paraformaldehyde (wt/vol). The brains were removed and processed for histology using standard techniques. For the electrophysiological experiments, we verified recording sites histologically with electrolytic lesions (15 s of 10 µA direct current across two wires of the same tetrode). For optogenetic experiments, virus expression was examined using a confocal microscope (Zeiss LSM 800). After histological verification, mice with incorrect virus injection or tetrode implantation were excluded from data analysis.

## Data analysis

All analyses were performed with MATLAB (Mathworks) and R (http://www.r-project.org/). All data are presented as mean ± SEM unless reported otherwise. All statistical tests were two-sided. For nonparametric tests, the Wilcoxon rank-sum test was used, unless data were paired, in which case the Wilcoxon signed-rank was used. To estimate neuronal learning rates, we used logistic functions of the form $f(x) = \frac{L}{1 + e^{-k(x-x_0)}}$. Learning rates are estimates of the $k$ parameter.

## Acknowledgements

We thank T Shelley for machining, and G Schoenbaum, M Pignatelli, and the Cohen Lab for comments. This work was supported by Life Sciences Research Foundation (FL), F30MH110084 (BAB), Klingenstein-Simons, MQ, NARSAD, Whitehall, R01DA042038, and R01NS104834 (JYC), and P30NS050274.

## Additional information

### Funding

| Funder | Grant reference number | Author |
| --- | --- | --- |
| National Institute on Drug Abuse | R01DA042038 | Jeremiah Y Cohen |
| National Institute of Neurological Disorders and Stroke | R01NS104834 | Jeremiah Y Cohen |
| Esther A. and Joseph Klingenstein Fund | | Jeremiah Y Cohen |
| MQ | | Jeremiah Y Cohen |
| Brain and Behavior Research Foundation | | Jeremiah Y Cohen |
| Whitehall Foundation | | Jeremiah Y Cohen |

| National Institute of Mental Health | F30MH110084 | Bilal A Bari |
|---|---|---|
| Life Sciences Research Foundation | | Federica Lucantonio |
| National Institute of Neurological Disorders and Stroke | P30NS050274 | Jeremiah Y Cohen |

The funders had no role in study design, data collection and interpretation, or the decision to submit the work for publication.

### Author contributions

Federica Lucantonio, Conceptualization, Formal analysis, Investigation, Writing - original draft, Writing - review and editing; Eunyoung Kim, Zhixiao Su, Anna J Chang, Bilal A Bari, Investigation; Jeremiah Y Cohen, Conceptualization, Resources, Formal analysis, Supervision, Funding acquisition, Methodology, Writing - original draft, Writing - review and editing

### Author ORCIDs

Jeremiah Y Cohen  https://orcid.org/0000-0002-4768-7124

### Ethics

Animal experimentation: All testing was conducted in accordance with the National Institutes of Health Guide for the Care and Use of Laboratory Animals and approved by the Johns Hopkins University Animal Care and Use Committee (protocol MO19M424).

### Decision letter and Author response

Decision letter https://doi.org/10.7554/eLife.57634.sa1
Author response https://doi.org/10.7554/eLife.57634.sa2

## Additional files

### Supplementary files

• Transparent reporting form

### Data availability

Data have been deposited on Dryad.

The following dataset was generated:

| Author(s) | Year | Dataset title | Dataset URL | Database and Identifier |
|---|---|---|---|---|
| Cohen JY | 2020 | Punishment history biases corticothalamic responses to motivationally-significant stimuli | https://doi.org/10.5061/dryad.hdr7sqvf5 | Dryad Digital Repository, 10.5061/dryad.hdr7sqvf5 |

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
