## [Decision Letter]

**Acceptance summary:**

A history of aversive experiences biases decision making, but the neural mechanisms through which this occurs is unclear. Using an odor guided go, no-go task, Lucantonio et al. show that when rodents have previously experienced aversive stimuli, they are more cautious and exhibit reduced reward seeking behavior. Importantly, this is accompanied by increased activity in neurons in the medial prefrontal cortex which project to the paraventricular nucleus of the thalamus and optogenetically increasing the activity of these cells mimics the effect of prior stress.

**Decision letter after peer review:**

Thank you for submitting your article "Punishment history biases corticothalamic responses to motivationally-significant stimuli" for consideration by *eLife*. Your article has been reviewed by 3 peer reviewers, one of whom is a member of our Board of Reviewing Editors, and the evaluation has been overseen by Michael Frank as the Senior Editor. The following individuals involved in review of your submission have agreed to reveal their identity: Mario Penzo (Reviewer #2); James Otis (Reviewer #3).

The reviewers have discussed the reviews with one another and the Reviewing Editor has drafted this decision to help you prepare a revised submission.

As the editors have judged that your manuscript is of interest, but as described below that additional experiments are required before a decision is reached, we would like to draw your attention to changes in our revision policy that we have made in response to COVID-19 (https://elifesciences.org/articles/57162). First, because many researchers have temporarily lost access to the labs, we will give authors as much time as they need to submit revised manuscripts. We are also offering, if you choose, to post the manuscript to bioRxiv (if it is not already there) along with this decision letter and a formal designation that the manuscript is "in revision at *eLife*". Please let us know if you would like to pursue this option. (If your work is more suitable for medRxiv, you will need to post the preprint yourself, as the mechanisms for us to do so are still in development.)

Summary:

The authors use electrophysiological and optogenetic techniques to study how prior aversive experiences alter olfactory cue based decision making. They report that when an aversive stimulus (airpuff) is added to the training regimen the resulting decision making behavior to other cues is more cautious and animals don't seek rewards as readily. This is accompanied by an increase in cue-evoked responding in medial prefrontal cortex (mPFC), and paraventricular thalamus (PVT) projecting mPFC, neurons. Finally, they show that optogenetic stimulation mPFC projections to PVT reduced cue-evoked reward seeking.

Essential revisions:

While all of the Reviewers thought that there were some potentially interesting results they felt that the analyses were seriously under-described and, in some cases, did not support the claims of the paper. Furthermore, there was a consensus opinion that the addition of a loss of function study was necessary to (a) go beyond what is already known from the literature, (b) test whether this pathway is important for prior aversive history to influence/inhibit reward seeking behavior and (c) to demonstrate that the projections to PVT, and not collateral projections to other brain regions, are important. Details of these and other concerns are outlined below.

1) The paper requires a loss of function study to adequately test a central claim of the paper that this pathway is essential for prior aversive history to influence/inhibit reward seeking behavior. The stimulation effects they see are a replication of previous results (Otis et al., 2017; Otis et al., 2019) and these effects are difficult to ascribe to prior history effects because optogenetic excitation may be impacting behavioral performance via mechanisms not related to punishment history. Ideally this would be a terminal inhibition experiment as the optogenetic terminal stimulation experiment leaves open the possibility of antidromic activitation of cell bodies and axon collaterals, resulting in effects which are not specific to projections to PVT.

2) A major issue with this paper is the way in which the results of statistical tests are presented. For example, in most cases the authors refer to overall ANOVA effects, but no group comparison tests are included. Specifically, on lines 258-262 the authors say that cue evoked firing rates were higher in cells from animals exposed to airpuff, statistics are provided and Figure 3C-D is referred to. They mention that they used a poisson regression to analyze this data, but very little information is given about this analysis, how it was used and the justification for its use. Furthermore, there is no statistical analysis which corresponds to the data presented in Figure 3C-D (this is also true for 1D and 5D). Figures reflecting the statistics with some kind of posthoc tests should be included so that the reader can get an idea of what odor mixtures this occurs in.

3) The authors want to emphasize the specific function of mPFC-PVT projection by measuring the different firing patterns of mPFC-PVT neurons with and w/o air-puff stressor. However, how specific this change is to this projection is unclear and this could simply reflect a general change occurring across most mPFC cells. They could do this by examining population tendencies across all mPFC cells and compare this to mPFC-PVT projection neurons.

4) Based on the figure legends and information in the figures, it is very difficult to understand what is represented in Figure 3B and the right panels in Figure 5A and B. These figures should be explained in the legend and explained/referred to in the text.

5) Also related to data presentation. While auROC is certainly a valuable way to identify neurons that can discriminate between reward and aversive cues, my understanding is that this is essentially a normalization and at such it limits appreciation of when a cue leads to an excitatory or an inhibitory response. It would be useful to also see the raw firing rate data for the neurons recorded. Also, for visualization purposes the authors should include pie charts summarizing electrophysiological data. This will help assess the data better and the actual proportion of cells with excitatory and inhibitory responses as well as cue discrimination.

6) In Figure 2 the auROC compares mPFC neuronal activity during sucrose vs denatonium trials rather than responses to each odor per se. Considering this analysis, technically a neuron could show greater inhibition to denatonium and no response to sucrose, and this would be completely hidden by the current analyses and figures. Data should be shown in the rawest form possible. One idea is that the authors could show spike frequency heatplots in addition to the existing auROC heatplots.

7) Description of data for Figure 4E-G indicate that 43% of neurons increase or decrease their activity in response to the sucrose-predictive odor, however the number that increase and the number that decrease in response to each odor is not indicated. This should be rather obvious to the reader and within both the figures and analysis.

8) To give the reader a better understanding of the dynamics of the cue response during the probe trials population/trial averaged PSTHs (similar to Figure 2B) should be presented for each stimulus condition in Figure 3C and 5C. Similarly, in Figure 3C and 5C, cue evoked firing rates are presented, but it is not clear how these were binned. Do they include only cue period, trace only, cue and trace, pre/post-stimulus? Finally, for Figure 1E, 3C and 5C it is unclear what is represented on the y-axis. Are these average values? This should be clarified.

9) Related to all electrophysiological figures, simply showing examples and then an average of all neurons does not take advantage of the powerful data collection that the authors have completed, could be quite misleading to readers, and is particularly important to consider given the heterogeneity of cue responses in the PFC-PVT pathway (shown here, also shown in Otis et al., 2017; Otis et al., 2019). I highly recommend inserting "example" right on the figure panel when it is an example, and "population average" when appropriate. Finally, to show the full and powerful datasets collected, authors should consider plotting and comparing cue responses for every recorded neuron. For example, for Figure 4 authors could plot sucrose odor vs baseline auROC for each cell on one axis and denatonium odor vs baseline auROC for each cell on the opposing axis for both air puff and control mice.

10) The authors have framed their study as addressing the effect of "punishment history" on motivationally-relevant stimuli. However, at present, their experimental design does not seem to rule out the possibility that what the authors are actually evaluating here is not the effect of punishment history, but rather the response to motivational conflict. Have the authors taken any steps to ensure that context was not the main driver of the behavioral and electrophysiological changes detected? If mice are being tested in the same context in which air puffs were presented, then their decreases in reward seeking may result from this context-derived motivational conflict.

11) Also related to experimental design is the notion that ambiguity seems to serve no purpose here. With the exception of the near zero response to Odor B (bitter solution), mice response to parametrically-varying odor mixtures and to Odor A itself were similarly decreased for the air puff group. There is a sense that the authors initially predicted an effect of punishment history and ambiguity on response ratio. This may have been the case had mice been tested in a different context, or trained using a different behavioral task in which punishment was not directly associated with the reward (as seems to be the case based on the current experimental design).

12) The authors claim that behavior for the first and second probe tests were not different and as such the data was pooled together. They should include the comparisons for the first and second probe test in the supplementary material, as it would be useful to see this quantification.

13) For optogenetic antidromic identification of mPFC-PVT projection neurons, the authors find cells that respond with a range of latencies and with a range of fidelity. One issue with the antidromic approach, however, pertains to their mention that only neurons with latencies under 15 ms and at least 70% spike probability were labeled as corticothalamic. But looking at the data shown in Figure 4d it seems that two of the neurons that passed the collision test had spike probability below 70%. Furthermore, I don't think this approach rules out orthodromic activation as many cells were not tested for collision and it is not possible to test the ability of cells to follow very high frequency stimulation or to get as accurate a readout of latency as would be possible with electrical antidromic approaches for which these criteria were developed. These issues should be clarified and discussed.

14) For the PVT projection neurons (Figure 5D) the authors note that "…in both groups, firing rates of mPFC-PVT neurons for odor mixtures scaled with the proportion of the mixture that was the sucrose predicting odor…". They make similar claims about the mPFC population as a whole (Figure 3D) However, based on the data in these figures it appears that, except for the non-ambiguous cue predicting bitter outcome, the cue response in both airpuff and non-airpuff groups was fairly flat, showing that there was no proportional scaling of the response. The authors should show some statistical support for this claim or change the interpretation.

[Editors' note: further revisions were suggested prior to acceptance, as described below.]

Thank you for resubmitting your work entitled "Aversive stimuli bias corticothalamic responses to motivationally-significant cues" for further consideration by *eLife*. Your revised article has been reviewed by 3 reviewers, one of whom is a member of our Board of Reviewing Editors, and the evaluation has been overseen by Michael Frank as the Senior Editor.

The manuscript has been greatly improved but there are some remaining issues that need to be addressed/clarified, as outlined in the reviewer comments below:

*Reviewer #1:*

The authors have addressed my previous concerns.

*Reviewer #2:*

It is this reviewer's opinion that the authors have appropriately addressed the concerns raised during the initial submission. In particular, the loss of function experiment has significantly improved the quality of the study by demonstrating that inhibition of the mPFC-PVT pathway prevents the formation of negative bias. Similarly, the addition of missing statistical information has facilitated the assessment of the authors' findings.

*Reviewer #3:*

The revisions overall were good, although I have some residual concerns listed below.

On one hand, addition of the optogenetic inhibition experiment adds causal information beyond what is already known regarding PFC◊PVT involvement in reward seeking. Additional data analysis and figure presentation has improved the manuscript significantly. New Figure 2A is fantastic – although I would have liked to see this in the phototagging experiment as well.

On the other hand, below are some of my concerns that either remain unaddressed or I simply can't find the in-text revisions. It would be helpful if authors indicate where exactly they have addressed each concern (e.g., line number).

Concerns Not Addressed:

1. The number of mice used for Figure 3, Figure 4, and Figure 5 remains unreported. Of concern is Figure 5F, where only 10 neurons are represented. Please report how many mice these 10 neurons come from.

2. Authors suggest that nomenclature of s^-1^ (which is used for neuronal and behavioral data analysis) is now described in the text, but I cannot find it in the methods, results, or legends. Am I missing it? Why not include it in the figure legend, like mean +/- SEM?

Additional Concern

1. The authors mention within their resubmission that recordings from Figure 2 were across 10 conditioning sessions, so is it possible that the same neurons were represented 10 times within a single dataset? If so, how do the authors account for sampling the same neurons repeatedly? Additionally, do the data from earlier conditioning sessions look the same as in later conditioning sessions? Considering that learning occurred across conditioning, it would be surprising if they were the same based on previous findings showing learning dependent cue-reward response development in dmPFC neurons that are stably maintained after learning (see PMID 34184635; Figure 4B). Please clarify this for me or simply add text within the results or discussion to indicate this limitation of the current analysis.

---

## [Author Response]

Essential revisions:While all of the Reviewers thought that there were some potentially interesting results they felt that the analyses were seriously under-described and, in some cases, did not support the claims of the paper. Furthermore, there was a consensus opinion that the addition of a loss of function study was necessary to (a) go beyond what is already known from the literature, (b) test whether this pathway is important for prior aversive history to influence/inhibit reward seeking behavior and (c) to demonstrate that the projections to PVT, and not collateral projections to other brain regions, are important. Details of these and other concerns are outlined below.1) The paper requires a loss of function study to adequately test a central claim of the paper that this pathway is essential for prior aversive history to influence/inhibit reward seeking behavior. The stimulation effects they see are a replication of previous results (Otis et al., 2017; Otis et al., 2019) and these effects are difficult to ascribe to prior history effects because optogenetic excitation may be impacting behavioral performance via mechanisms not related to punishment history. Ideally this would be a terminal inhibition experiment as the optogenetic terminal stimulation experiment leaves open the possibility of antidromic activitation of cell bodies and axon collaterals, resulting in effects which are not specific to projections to PVT.

We agree with the reviewers that a loss of function experiment is crucial to support our conclusions. We added a new experiment in which we prevent the formation of negative bias by inhibiting mPFC→PVT axons during air-puff trials *only* during conditioning days (Figure 7 and Supplementary Figure 4). Mice receiving inhibition of mPFC axons over PVT during aversive cues had increased anticipatory licking to motivationally-relevant stimuli on probe trials on later days. There was no inactivation during probe trials. Thus, axon terminal inhibition during learning affected future behavior. This demonstrates that mPFC→PVT activity during learning affects future responses to motivationally-relevant stimuli.

2) A major issue with this paper is the way in which the results of statistical tests are presented. For example, in most cases the authors refer to overall ANOVA effects, but no group comparison tests are included. Specifically, on lines 258-262 the authors say that cue evoked firing rates were higher in cells from animals exposed to airpuff, statistics are provided and Figure 3C-D is referred to. They mention that they used a poisson regression to analyze this data, but very little information is given about this analysis, how it was used and the justification for its use.

We apologize for the lack of detail in motivating and describing statistical tests. We used a Poisson regression which tests for effects of categorical variables (in our case, air puff exposure) on variables generated from a discrete nonnegative probability distribution (in our case, spike counts).

Using this regression, we found that cue-evoked firing rates were significantly higher in mice exposed to air puffs (Figures 3C and 3D; odor mixture *z* = 0.18 ± 0.015, air-puff group *z* = 0.043 ± 0.014, stimulus-group interaction *z* = 0.033 ± 0.019, *p <* 0.001). As we also explain in the text, when we use the term “odor mixture,” we refer to all combinations of sucrose and denatonium proportions including pure denatonium (100/0 mixture) and pure sucrose (0/100 mixture). As suggested by the reviewers, we also added group comparison tests where appropriate (Tukey HSD or Wilcoxon rank sum tests). We conclude that firing rates were higher at each odor mixture in mice exposed to air puffs (Figure 3D, all *p <* 0.05). We clarified these analyses in the text.

Furthermore, there is no statistical analysis which corresponds to the data presented in Figure 3C-D (this is also true for 1D and 5D). Figures reflecting the statistics with some kind of posthoc tests should be included so that the reader can get an idea of what odor mixtures this occurs in.

As described above, we expanded our description of the Poisson regression test that was used for analyzing data in Figures 3C, 3D, and 5D, and we also ran post hoc tests to make specific comparisons.

3) The authors want to emphasize the specific function of mPFC-PVT projection by measuring the different firing patterns of mPFC-PVT neurons with and w/o air-puff stressor. However, how specific this change is to this projection is unclear and this could simply reflect a general change occurring across most mPFC cells. They could do this by examining population tendencies across all mPFC cells and compare this to mPFC-PVT projection neurons.

We thank the reviewers for raising this concern. We did not intend to draw conclusions about the specificity of mPFC→PVT neurons; indeed, we do not have experiments comparing these identified cells to cells with other projection targets. We have revised the text to make it clear.

4) Based on the figure legends and information in the figures, it is very difficult to understand what is represented in Figure 3B and the right panels in Figure 5A and B. These figures should be explained in the legend and explained/referred to in the text.

We apologize for the lack of clarity of those figure panels. We added more details to the legend.

5) Also related to data presentation. While auROC is certainly a valuable way to identify neurons that can discriminate between reward and aversive cues, my understanding is that this is essentially a normalization and at such it limits appreciation of when a cue leads to an excitatory or an inhibitory response. It would be useful to also see the raw firing rate data for the neurons recorded. Also, for visualization purposes the authors should include pie charts summarizing electrophysiological data. This will help assess the data better and the actual proportion of cells with excitatory and inhibitory responses as well as cue discrimination.

We thank the reviewers for this suggestion. We analyzed firing rate for each neuron in reward and aversive trials without any normalization. Those results are summarized in Figures 2A. Specifically, we compared firing rates of neurons for sucrose and denatonium trials in mice exposed and unexposed to air puffs relative to their pre-CS firing rates. Based on those results, there is a positive correlation between firing rates in response to sucrose-predicting and denatonium-predicting CS in mice exposed (*r* = 0.59 ± 0.04, 95% CI, *p <* 0.0001) and unexposed (*r* = 0.58 ± 0.04, 95% CI, *p <* 0.0001) to air puffs.

6) In Figure 2 the auROC compares mPFC neuronal activity during sucrose vs denatonium trials rather than responses to each odor per se. Considering this analysis, technically a neuron could show greater inhibition to denatonium and no response to sucrose, and this would be completely hidden by the current analyses and figures. Data should be shown in the rawest form possible. One idea is that the authors could show spike frequency heatplots in addition to the existing auROC heatplots.

As described in the response above, we analyzed raw firing rates for each neuron in reward and aversive trials without normalization. Those results are summarized in Figure 2A.

7) Description of data for Figure 4E-G indicate that 43% of neurons increase or decrease their activity in response to the sucrose-predictive odor, however the number that increase and the number that decrease in response to each odor is not indicated. This should be rather obvious to the reader and within both the figures and analysis.

Sorry for not having made this explicit. We added this information in the current version of the manuscript. To be clear, data presented in this figure are relative to air-puff-predicting cues.

8) To give the reader a better understanding of the dynamics of the cue response during the probe trials population/trial averaged PSTHs (similar to Figure 2B) should be presented for each stimulus condition in Figure 3C and 5C.

We agree with the reviewers that showing PSTHs will give the reader a better understanding of the dynamics of the cue response so we have now added these data in Figure 3E and 5E.

Similarly, in Figure 3C and 5C, cue evoked firing rates are presented, but it is not clear how these were binned. Do they include only cue period, trace only, cue and trace, pre/post-stimulus?

Cue-evoked firing rates were calculated during the cue and excluded the trace period. We clarified this in the text.

Finally, for Figure 1E, 3C and 5C it is unclear what is represented on the y-axis. Are these average values? This should be clarified.

Yes, those are average values. We clarified this in the text.

9) Related to all electrophysiological figures, simply showing examples and then an average of all neurons does not take advantage of the powerful data collection that the authors have completed, could be quite misleading to readers, and is particularly important to consider given the heterogeneity of cue responses in the PFC-PVT pathway (shown here, also shown in Otis et al., 2017; Otis et al., 2019). I highly recommend inserting "example" right on the figure panel when it is an example, and "population average" when appropriate.

Thank you for this suggestion. We have now made very clear in the text and in the figures where examples vs population averages are shown.

Finally, to show the full and powerful datasets collected, authors should consider plotting and comparing cue responses for every recorded neuron. For example, for Figure 4 authors could plot sucrose odor vs baseline auROC for each cell on one axis and denatonium odor vs baseline auROC for each cell on the opposing axis for both air puff and control mice.

Thank you for suggesting these analyses. As previously mentioned, we have analyzed and plotted firing rate for each neuron in reward and aversive trials and those results are summarized in Figure 2A. We compared firing rates of neurons for sucrose and denatonium trials in mice exposed and unexposed to air puffs relative to their pre-CS firing rates.

10) The authors have framed their study as addressing the effect of "punishment history" on motivationally-relevant stimuli. However, at present, their experimental design does not seem to rule out the possibility that what the authors are actually evaluating here is not the effect of punishment history, but rather the response to motivational conflict. Have the authors taken any steps to ensure that context was not the main driver of the behavioral and electrophysiological changes detected? If mice are being tested in the same context in which air puffs were presented, then their decreases in reward seeking may result from this context-derived motivational conflict.

We would like to thank the reviewers for this important point. We agree with the reviewers that mice being tested in the same context in which air puffs are presented may result in a conflict and the current experimental design does not allow to rule out an effect of context-derived motivational conflict on the obtained results. We have now added text to the Discussion about this issue.

11) Also related to experimental design is the notion that ambiguity seems to serve no purpose here. With the exception of the near zero response to Odor B (bitter solution), mice response to parametrically-varying odor mixtures and to Odor A itself were similarly decreased for the air puff group. There is a sense that the authors initially predicted an effect of punishment history and ambiguity on response ratio. This may have been the case had mice been tested in a different context, or trained using a different behavioral task in which punishment was not directly associated with the reward (as seems to be the case based on the current experimental design).

As for the previous point, we have discussed the methodological issue of having tested the mice in the same context in which the air puff has been presented in the current version of the manuscript. However, in our current experimental design, the punishment was never directly associated with the reward.

12) The authors claim that behavior for the first and second probe tests were not different and as such the data was pooled together. They should include the comparisons for the first and second probe test in the supplementary material, as it would be useful to see this quantification.

We agree with the reviewers that presenting the data for the first and second probe tests separately is important, so we have now added those results in Supplementary Figure 1.

13) For optogenetic antidromic identification of mPFC-PVT projection neurons, the authors find cells that respond with a range of latencies and with a range of fidelity. One issue with the antidromic approach, however, pertains to their mention that only neurons with latencies under 15 ms and at least 70% spike probability were labeled as corticothalamic. But looking at the data shown in Figure 4d it seems that two of the neurons that passed the collision test had spike probability below 70%. Furthermore, I don't think this approach rules out orthodromic activation as many cells were not tested for collision and it is not possible to test the ability of cells to follow very high frequency stimulation or to get as accurate a readout of latency as would be possible with electrical antidromic approaches for which these criteria were developed. These issues should be clarified and discussed.

We agree that neurons that were not subjected to collision tests may have exhibited orthodromic activation via local synapses. We revised the text to clarify this caveat. Regarding the criteria for identifying cells, we used relatively conservative criteria for identifying corticothalamic neurons. Although its use is still relatively new, we used stimulation frequencies in the range of other studies that used collision tests with excitatory opsins to identify corticothalamic and corticostriatal neurons (Economo et al., 2018; Bari et al., 2019).

14) For the PVT projection neurons (Figure 5D) the authors note that "…in both groups, firing rates of mPFC-PVT neurons for odor mixtures scaled with the proportion of the mixture that was the sucrose predicting odor…". They make similar claims about the mPFC population as a whole (Figure 3D) However, based on the data in these figures it appears that, except for the non-ambiguous cue predicting bitter outcome, the cue response in both airpuff and non-airpuff groups was fairly flat, showing that there was no proportional scaling of the response. The authors should show some statistical support for this claim or change the interpretation.

Thank you for the comment. We agree with the reviewer and we have changed the interpretation of the results.

[Editors' note: further revisions were suggested prior to acceptance, as described below.]

Reviewer #3:The revisions overall were good, although I have some residual concerns listed below.On one hand, addition of the optogenetic inhibition experiment adds causal information beyond what is already known regarding PFC◊PVT involvement in reward seeking. Additional data analysis and figure presentation has improved the manuscript significantly. New Figure 2A is fantastic – although I would have liked to see this in the phototagging experiment as well.On the other hand, below are some of my concerns that either remain unaddressed or I simply can't find the in-text revisions. It would be helpful if authors indicate where exactly they have addressed each concern (e.g., line number).Concerns Not Addressed:1. The number of mice used for Figure 3, Figure 4, and Figure 5 remains unreported. Of concern is Figure 5F, where only 10 neurons are represented. Please report how many mice these 10 neurons come from.

Thank you for this comment and sorry for the confusion. The number of mice for Figure 3 are reported in line 143 (12 mice, 5 exposed to air puffs, 7 unexposed). The number of mice for Figures 4 and 5 are reported in the caption of Figure 4—figure supplement 1 (9 mice, 3 unexposed and 6 exposed to air puff). The 10 neurons in Figure 5F are from the 6 mice exposed to air puffs.

2. Authors suggest that nomenclature of s^-1^ (which is used for neuronal and behavioral data analysis) is now described in the text, but I cannot find it in the methods, results, or legends. Am I missing it? Why not include it in the figure legend, like mean +/- SEM?

If we understand correctly, the reviewer is asking about the “s^−1^” notation, which we used to abbreviate “per second.”

Additional Concern1. The authors mention within their resubmission that recordings from Figure 2 were across 10 conditioning sessions, so is it possible that the same neurons were represented 10 times within a single dataset? If so, how do the authors account for sampling the same neurons repeatedly? Additionally, do the data from earlier conditioning sessions look the same as in later conditioning sessions? Considering that learning occurred across conditioning, it would be surprising if they were the same based on previous findings showing learning dependent cue-reward response development in dmPFC neurons that are stably maintained after learning (see PMID 34184635; Figure 4B). Please clarify this for me or simply add text within the results or discussion to indicate this limitation of the current analysis.

Thank you for this comment. During conditioning sessions, tetrodes were advanced 40–80 *µ*m at the end of the session when fewer than 4 tetrodes had single units. It is highly unlikely that the same neuron is included multiple times in the data set. To clarify this point in the text, we added a sentence in the methods section in which we specify that we cannot completely rule out that a specific neuron may be represented more than one time in the dataset (lines 472–473).